# Nurses’ Attitudes and Factors Affecting Use of Electronic Health Record in Saudi Arabia

**DOI:** 10.3390/healthcare11172393

**Published:** 2023-08-25

**Authors:** Awatif M. Alrasheeday, Bushra Alshammari, Sameer A. Alkubati, Eddieson Pasay-an, Monirah Albloushi, Awayed M. Alshammari

**Affiliations:** 1Nursing Administration Department, College of Nursing, University of Hail, Hail 2440, Saudi Arabia; a.alrasheeday@uoh.edu.sa; 2Medical Surgical Nursing Department, College of Nursing, University of Hail, Hail 2440, Saudi Arabia; bu.alshammari@uoh.edu.sa; 3Department of Nursing, Faculty of Medicine and Health Sciences, Hodeida University, Hodeida P.O. Box 3114, Yemen; 4Maternal and Child Nursing Department, College of Nursing, University of Hail, Hail 2440, Saudi Arabia; e.pasayan@uoh.edu.sa; 5Medical Surgical Department, College of Nursing, King Saud University, Riyadh 11451, Saudi Arabia; malbloushi@ksu.edu.sa; 6Nursing Administration, King Khalid General Hospital, Ministry of Health, Hafar Al Batin 39921, Saudi Arabia; awalshamari@moh.gov.sa

**Keywords:** electronic health records, nurses, attitudes, Saudi Arabia

## Abstract

(1) Background: Nurses’ attitudes toward electronic health records (EHRs) is a very valuable issue that needs to be evaluated, understood, and considered one of the main factors that can lead to its improvement or handicap its implementation. This study aimed to assess nurses’ attitudes toward EHRs and associated factors that affect the implementation of EHRs in different hospitals in Saudi Arabia. (2) Methods: A cross-sectional study was utilized to collect data from 297 nurses working in public hospitals and primary healthcare centers in Ha’il Province from January to May 2023. Data were collected using the Nurses’ Attitudes Towards Computerization questionnaire and a sociodemographic and work-related characteristics sheet. (3) Results: Most of the participants’ attitude scores (81.1%, *n* = 241) were more than or equal to 60, representing positive attitudes, whereas 18.9% (*n* = 56) of the nurses’ scores were less than 60, which is interpreted as negative attitudes. There was a significant relationship between nurses’ attitudes toward EHRs and a participants’ sex, where males had a more positive attitude than females (*p* < 0.001). Particularly, young nurses and those who had previous computer experience had a more positive attitude than older nurses and those who had no computer experience (*p* = 0.044 and < 0.001, respectively). Saudi nurses holding a master’s degree had significantly more positive attitudes toward EHRs than non-Saudi nurses holding a bachelor’s or diploma degree (*p* = 0.007 and 0.048, respectively). Nurses with less experience (less than five years) in the nursing field had a significantly positive attitude. Multiple linear regression showed that sex (*p* = 0.038), level of education (*p* = 0.001), and previous computer experience (*p* < 0.001) were independent factors of nurses’ knowledge of EHRs. (4) Conclusion: The majority of nurses had positive overall attitudes toward using EHRs. Nurses who are Saudi nationals, male, younger, have previous computer experience, and have less than five years of experience had a more positive attitude toward EHRs than nurses who are non-Saudi, female, older, have no computer experience, have bachelor’s or diploma degree, and have less than five years of experience, respectively. Sex, education level, and previous computer experience were independent factors of nurses’ knowledge of EHRs.

## 1. Introduction

Electronic health records (EHRs) are electronic records of health-related information that are used to save and retrieve patients’ information [1]. EHRs enable medical staff to access patient data from any location at all times and perform other duties, such as scheduling, issuing requests, seeing laboratory and radiology findings, and updating and prescribing clinical notes [2]. There are numerous advantages to using EHRs, such as improving the quality of medical and nursing documentation, which can lead to a decrease in the occurrence of medical errors related to misinterpretation and, consequently, improve medical and nursing care [3]. Nurses have to improve their abilities for developing and implementing documentation quality to achieve the best patient safety and health outcomes [4].

Nurses’ attitudes toward EHRs is an important issue that needs to be evaluated, understood, and considered one of the main factors that can lead to its improvement or handicap its implementation [5,6]. Therefore, the improvement and strength of EHRs depends mainly on nurses accepting their use, which can provide stimulation for their continuous learning and training to use them [5,7].

EHRs were first implemented by the Saudi Arabian government in some healthcare facilities a few decades ago [3,6]. On a national level, however, the adoption of these systems has moved slowly [6] and their acceptance and implementation has tended to lag because of obstacles [8]. This slow adoption may be due to a lack of computer expertise, expensive costs, security concerns, workflow challenges, and time [9]. Saudi Arabia is putting significant of effort into providing patients with modern healthcare [10]. Once a patient’s medical history is represented electronically, it may include crucial administrative, clinical, laboratory, and radiological data [11], making it easier for different healthcare professionals to save and share valuable health data [12]. It enhances patient care by enhancing the accuracy of medical records and lowering the risk of medical error [13] as well as improving healthcare professionals’ ability to care for patients [14,15].

Numerous investigations have attempted to link the adaptability of clinicians to electronic record systems that necessitate prior computer knowledge from using the EHR system in Saudi Arabian healthcare facilities as well as encouraging changes in health care workers’ attitudes toward it [16,17]. However, as part of efforts to boost the efficiency with which primary healthcare centers (PHCs) manage chronic diseases, the Saudi Ministry of Health (MoH) installed an EHR system in every PHC [18]. The MOH’s National E-Health Strategy was created to make it easier for the healthcare sector to transition from a paper-based to an electronic platform; this is to improve the quality of healthcare services and to realize its proactive vision for e-health: a “safe, efficient health system, based on the care centered on a patient, standard-oriented, and supported by the e-health” is what has motivated the adoption of EHRs in Saudi Arabia [19].

Despite the advantages of using the EHR system in medical and healthcare operations, adopting such systems is still slow and encounters resistance from healthcare professionals [9]. Nurses’ opinions and attitudes towards the use of EHRs is very important factor that should be investigated to improve their adherence to use such technology in their practice [5]. Recently a study demonstrated nurses’ positive attitudes to the use of EHRs in Jordan [20]. The health care workers reported acceptable attitude and satisfaction towards using EMRs [1]. However, in a systematic review study that illustrated the barriers and obstacles affecting the use of EHRs in Saudi Arabia, such as inadequate computer knowledge, a lack of system customization to hospital requirements, and improper training by the information technology teams [6]. These obstacles may be social, managerial, organizational, or even political [15,21]. There is limited research regarding nurses’ attitudes toward using EHRs, different factors that can impact the implementation of EHRs, and challenges that nurses face when implementing EHR systems. Thus, this present study was designed to assess nurses’ attitudes toward EHRs and the associated factors that affect the implementation of EHRs in different hospitals in Saudi Arabia. The results may be used to determine gaps and interventional plans that could help the nurses improve their practice toward EHRs. In addition, the results may be used as a basis for increasing the awareness of the healthcare administrators to arrange for more workshops and training programs to improve nurses’ attitudes toward EHRs. The uniqueness of this study is that the data were taken from the main hospitals of one of the biggest regions of Saudi Arabia.

## 2. Materials and Methods

### 2.1. Design and Setting

Because this study entails employing many groups of people that differ in the variable of interest but share other characteristics (such as socioeconomic level, educational background), a cross-sectional study was used [22]. Cross-sectional surveys were used by the researchers because they allow for the simultaneous collection of data on prevalence and outcome, as with a demographic factor. The data were collected from nurses working in all public hospitals that provided health services for patients in and outside Hail city, Saudia Arabia, namely, (1) King Salman Hospital, (2) Sharaf Hospital, (3) King Khaled Hospital, and (4) Hail General Hospital. In addition, all PHCs (28 center) in that area were included in this study. The hospitals and PHCs, with different specialties clinics, provide continuous health care services for the population and their dependents. The study was conducted from January to May 2023.

### 2.2. Sample

Nurses were included if they had dealt with EHRs for at least one year or more and if they agreed to participate. Nurses were excluded from the study if they had dealt with EHRs for less than one year, if they had only associate, practical or training roles, and if they worked only in supportive services, such as laboratory and radiology departments. Using OpenEpi, Version 3.01 (www.openepi.com, accessed on 12 November 2022), a software used to determine the sample size for a Proportion or Descriptive Study by to calculating the sample size necessary to determine the frequency of a factor in a population [23], a minimum sample size of 292 nurses was estimated based on a population size of 1200, a 95% confidence level, and 5% absolute precision [24]. A total of 400 nurses were given the questionnaire, and 297 responded, representing a response rate of 74.25%. The nurses were selected from the study population by convenience sampling, one of the nonprobability sampling methods, as it is considered the most economical, easiest, as well as fastest way to collect data from a population [25,26].

### 2.3. Instrument

The Nurses’ Attitudes Towards Computerization (NATC) questionnaire, developed by Stronge and Brodt (1985) [27], was utilized in this study and is considered an effective tool to measure the attitudes of nurse towards computerization and EHRs [5,20,28]. This questionnaire consisted of 20 items with a 5-point Likert scale, which ranged from strongly agree to strongly disagree. The total scores ranged from 20 to 100. Scores of 60 or more were considered positive attitudes, while scores less than 60 were considered negative attitudes [20,28]. The questionnaire included five domains: benefits to the institution (4 items), patient care quality (6 items), superior capabilities of computers (6 items), willingness to use computers (1 item), and legal issues in computer use (3 items). These domains reflect what the nurses thought to be the major concerns related to computer use [29]. Also, nurses’ sociodemographic and profession-related characteristics (e.g., sex, age, educational level, nationality, years of experience, working unit, and position) were included. The content validity and reliability of the questionnaire was evaluated in previous studies [28,29,30]. To test the reliability of the questionnaire, a pilot study was conducted with 20 nurses (who were excluded from the study sample). The reliability analysis revealed internal consistency with the Cronbach’s alpha coefficient (0.919), which means that it is reliable. The questionnaire took nurses about 15–20 min to complete.

### 2.4. Data Collection

Nurses were approached and invited by the researchers to participate in this study. After explaining the study’s purpose, a researcher distributed the questionnaires along with informed consent forms to the nurses during their break time and waited until they filled out the questionnaires. The English version of the questionnaire was used in this study.

### 2.5. Ethical Considerations

This study was approved by the Research Ethics Committee of the University of Hail, Saudi Arabia (Ethical Approval No: H-2023-168). Each participant signed a consent form after being informed about the study’s objectives. Participants were informed that their participation was completely voluntary, with no incentives given to participants to complete the survey, and that they could discontinue it at any moment. Data confidentiality and respondent anonymity were guaranteed.

### 2.6. Data Analysis

The data were analyzed using IBM SPSS Statistics software, Version 27 (IBM Corp., Armonk, NY, USA) [31,32]. Normally distributed continuous variables were described as mean and standard deviation (SD), and categorical variables were described as frequencies and percentages. The Kolmogorov–Smirnov test was used to determine the normality of distribution [33]. The results indicate that the *p*-value was less than 0.05, which means that the data were nonnormally distributed. Accordingly, nonparametric statistics (Mann–Whitney test or Kruskal–Wallis tests) were used in this study to investigate the relationship between the independent variables and the total scores of the attitudes [34]. Factors significantly associated with nurses’ attitudes toward EHRs were further analyzed using multiple linear regression. Statistical significance was set at *p* < 0.05.

## 3. Results

### 3.1. Participants’ Sociodemographic and Work-Related Characteristics

Table 1 shows that the majority of nurses are female (68.7%) and aged between 31 and 40 years. More than half of the nurses (58.6%) have bachelor’s degrees, less than two-thirds (63.3%) are Saudi, and more than one-third (37.7%) had 5–10 years of experience. Most of the nurses had previous computer experience (79.1%), more than one-third (36%) were working in intensive care units, and more than half (58.9%) were working as staff nurses.

### 3.2. Nurses’ Attitudes toward EHRs

As shown in Table 2, nurses’ overall attitudes toward using EHRs were positive. A descriptive analysis of the total attitude scores revealed that the mean (SD) was 70.33 (12.83), the minimum score was 29, and the maximum score was 100. Also, most of the participants’ attitude scores (81.1%, *n* = 241) were more than or equal to 60, representing positive attitudes, whereas 18.9% (*n* = 56) of the nurses’ total scores were less than 60, which is interpreted as negative attitudes. The items “Paperwork for nurses has been greatly reduced by the use of computers”, “Computers save steps and allow the nursing staff to become more efficient”, and “Computers make nurses’ jobs easier” were ranked higher than others, with means 4.00, 3.95, and 3.94, respectively. The items “Computers should only be used in the financial department” and “If I had my way, nurses would never have to use computers” scored the lowest, with means 2.97 and 2.94, respectively.

### 3.3. Relationship between Nurses’ Sociodemographic and Work-Related Factors and Their Attitudes toward Ehrs

Table 3 illustrates the relationship between nurses’ attitudes toward using EHRs and their sociodemographic and work-related variables. There was a significant relationship between nurses’ attitudes toward EHRs and their sex, with males having more positive attitudes than females (*p* < 0.001). Young nurses and those who had previous computer experience had significantly more positive attitudes than older nurses and those without computer experience (*p* = 0.044 and < 0.001, respectively). Also, Saudi nurses holding a master’s degree had a significantly more positive attitude toward EHRs than non-Saudi nurses holding a bachelor’s or diploma degree (*p* = 0.007 and 0.048, respectively). Nurses with less experience (less than five years) in the nursing field had a significantly positive attitude. Nurses’ positions and their units had no significant relationship with nurses’ attitudes regarding EHRs (*p* > 0.05).

### 3.4. Independent Factors of Nurses’ Attitudes toward EHRs

The multiple linear regression showed that sex (*p* = 0.038), level of education (*p* = 0.001), and previous computer experience (*p* < 0.001) were the independent factors of nurses’ knowledge of EHRs (see Table 4).

## 4. Discussion

This study aimed to assess nurses’ attitudes toward EHRs and the associated factors that affect the implementation of EHRs in different hospitals in Saudi Arabia. In this current study, 81.1% of the nurses had overall positive attitudes toward using EHRs, which implies that the nurses regarded EHRs favorably and were willing to incorporate them into their daily operations. In healthcare settings, nurses are among the largest prospective users of electronic medical record systems [35]. Based on the results of an attitude questionnaire, most of these nurses agreed that computerized documentation is necessary and acceptable, similar to a Palestinian study [5]. Moreover, nurses in Jordan have embraced EHRs due to their perceived utility and ease of use [36]. Nurses who said that the EHR system helped them do better in their clinical nursing jobs were more likely to think that the EHR system helped them do better in their clinical nursing careers [37,38]. Additionally, positive insights were expressed by nurses in Australia about adoption of the EMR in their workplace environment as a results of their expectations that EMR may assist them in significant ways, such as timely and legible documentation, and to provide assistance in improving patient safety and care delivery [39]. Conversely, 18.9% of the nurses had a negative attitude, which implies that EHRs were met with resistance or reluctance from nurses.

In 2016, nurses were unsatisfied with the existing implementation of nursing capabilities in EHRs [40]. The authors showed that over half of the comments pointed to problems with the system itself (e.g., poor system usability, nonintegrated systems, poor interoperability, a lack of standards, and limited functionality/missing components), followed by problems with user tasks (e.g., the failure of systems to meet nursing clinical needs and non-nursing-specific systems), and problems with their working conditions (e.g., the low prevalence of EHRs, and a lack of user training). In addition, in a study conducted in a State Tertiary Hospital in Southwest Nigeria to investigate healthcare workers’ computer skills during the uptake of EHRs, researchers found that the majority of the participants did not have sufficient skills in Microsoft Excel and Microsoft Access [41]. Other barriers were reported by a systematic review study, such as inappropriate resources, training, and technical and educational support, as well as poor skills and literacy of healthcare providers [15].

The items “Paperwork for nurses has been greatly reduced by the use of computers”, “Computers save steps and allow the nursing staff to become more efficient”, “Computers make nurses’ jobs easier”, and “Computerization of nursing data offers nurses a remarkable opportunity to improve patient care” had higher means, which implies that the majority of nurses’ view computerization positively for their work and patient care. It was found that nurses generally viewed computers in a positive light. Nursing care can be made safer, more efficient, and more effective through the use of EHRs [42].

According to Lindén-Lahti et al. (2022), the shift from paper to digital documentation has overarching goals: to better promote communication, reduce errors, and ensure that patients receive consistent care [43]. The authors concluded that nursing records should be improved in a way that takes into account nurses’ wants and needs, the effects of any new capability on existing workflows, and consistency with domain-specific representation models of standardized data.

The items “Computers should only be used in the financial department” and “If I had my way, nurses would never have to use computers” scored the lowest means, which implies that some nurses have a negative view of computerization and would rather not interact with computers. This can be a result of not being exposed to technology. As shown in one study, most nurses in Lesotho do not have adequate computer skills, which is strongly correlated with the length of time that has passed since they last obtained a qualification and with a lack of exposure to computers [44]. Nurses’ willingness to adopt computerization is indicative of their understanding of the merits of using technology in their profession [41]. This willingness of nurses to adopt EHR means that they are aware of the positive effects that innovations and informatics can have on high-quality nursing practice and how this impacts the patients’ health [39]. Being open to change and willing to embrace computerization are both traits that are crucial in the rapidly evolving world of healthcare technologies today [45].

An increase in the use of digital technologies, such as electronic health records and clinical decision support systems, can enhance the quality and effectiveness of patient care if this is recognized [46]. The readiness to embrace computerization is also indicative of an openness to change, which is especially important in the current era of quickly developing healthcare technologies [41,46]. From a pedagogical standpoint, nursing schools should make it a top priority to teach students how to use computers and the Internet effectively. Nursing education can help educate future nurses to succeed in a healthcare system that is more reliant on information technology by emphasizing the importance of computer literacy.

It was found that there was a significant relationship between nurses’ attitudes toward EHRs and sex, as males rated more positive attitudes than females. This means that there is a significant sex disparity in nurses’ perspectives on EHRs, with male nurses generally holding a more positive opinion of these systems. Being a male was also a major factor in provider preparedness, lending credence to meet the idea that men are more comfortable and enthusiastic about using technology than women [47]. Various sociocultural issues, such as historical sex roles, preconceptions, and biases, contribute to meeting the sex gap in nurses’ adoption and employment of technology [48]. Educational and occupational prospects, access to technology, and societal norms about technology use may all be affected by these variables. If we want to foster an environment where people of all sex can succeed in the field of technology, we must question and combat such prejudices and biases. To close the gap and allow everyone (regardless of sex) an opportunity to succeed based on their interests and skills, we must promote equal opportunities, provide education and training, and address institutional hurdles [49].

Young nurses and those who had previous computer experience had more positive attitudes than older nurses who had no computer experience, which means that age and previous experience with computers are two crucial characteristics that can influence nurses’ attitudes toward utilizing technology within the healthcare industry. As disclosed in Huber and Schubert’s (2019) [50] study, younger nurses and those with prior computer experience were more optimistic than senior nurses and those with prior computer experience. Nurses’ perceptions of the value of information systems are moderated by their level of computer literacy [51].

Positive attitudes toward EHRs can catalyze the promotion of their use among colleagues, thereby encouraging the widespread adoption and integration of digital technologies in nursing practices. Young nurses and those with computer experience can play a crucial role in educating their colleagues on EHR systems, thereby facilitating seamless transitions and reducing change resistance [52]. Additionally, utilizing these people’s positive attitudes and current digital competencies might help nursing administrators to understand that the early training of EHR ensures that future nurses are ready to accept and use technology in their professional positions [53]. It is also crucial for nursing leaders to ensure that all nurses, regardless of age or technological experience, receive adequate training and support in the use of EHRs.

Also, Saudi nurses and those holding master’s degrees had significantly more positive attitudes toward EHRs than non-Saudi nurses and those holding bachelor’s or diploma degrees, which means that nurses’ perspectives and attitudes about EHRs are significantly influenced by their nationality and level of education. According to a study by Strudwick et al. (2015), countries located in the Middle East saw a surge in the installation and implementation of such technologies, with nurses making up the largest user group. The findings of a study carried out in Taiwan provide further evidence that differences in national culture have a substantial influence on the behavioral intention of nurses to use EHRs [54]. Furthermore, in Turkey’s primary care settings, healthcare professionals with master’s degrees were more likely to favor EHRs [55]. This study shows that nurses, especially those older and without master’s degrees, need EHR-specific training and instruction. These training programs should train nurses to make better use of EHRs by making them more familiar with their features and benefits.

Nurses who had less experience (less than five years) in the nursing field had a significantly positive attitude, which means that newer generations of nurses are more responsive and flexible because they are better trained when it comes to integrating new technologies into healthcare environments. Topaz et al. (2016) [40] found that novice nurses held a favorable view of EHRs. In the same view, nursing graduates who had average innovation ratings that were higher than average were more likely to engage in innovative behaviors [56], and younger faculty members in nursing were more likely to have favorable attitudes toward increased technology use and adoption [49]. This study established that nurses who have a positive stance toward EHRs are more likely to make good use of them in their daily work. They are more likely to embrace change, learn how to use new tools, and make full use of EHR systems to improve patient care. This optimistic perspective on EHRs can boost documentation quality, facilitate better inter-professional communication, and ultimately benefit patients [15]. Furthermore, these nurses can promote the use of EHRs among their colleagues, fostering a digital transformation culture in healthcare facilities.

Nurses’ positions and their working units have no significant relationship with nurses’ attitudes regarding EHRs, which means that the disposition of nurses toward EHRs does not vary with rank or department. That is to say, nurses’ perspectives toward EHRs are consistent, whether they work in the intensive care unit, the emergency room, as registered nurses, or as nurse practitioners. The attitudes of nurses towards EHRs were found to be consistent across ranks and units in the research conducted at Sri Ramakrishna Hospital [57]. Consistent attitudes across ranks and units suggest that nurses are more likely to collaborate and support the implementation of EHRs if they hold consistent perspectives [58]. Working toward the successful adoption of the system, they will likely have a unified voice and approach. It is simpler to establish standardized protocols and best practices for documentation, data entry, and communication using the EHR system when nurses share similar perspectives. As a result, they will be better able to work together and coordinate patient care by understanding one another’s difficulties, requirements, and preferences when it comes to EHRs. All nurses, regardless of position or department, can benefit from individualized training programs.

A multiple linear regression showed that sex, level of education, and previous computer experience were independent factors of nurses’ knowledge of EHRs. This means that when determining nurses’ understanding of EHRs, these three characteristics are treated individually and independently from one another. There was no discernible difference in attitudes regarding the acceptance of EHRs between male and female nursing staff members working in public hospitals in Palestine [5]. However, it is possible to anticipate the acceptability of EHRs among nursing employees by looking at user demographic factors, health care facilities’ upper-management staff, IT support, and EHR system quality [59]. In a study conducted in Jordan [36], nurses’ willingness to adopt EHRs was found to be positively connected with their perceptions of the technology’s utility and convenience of use. It was observed that nurses’ prior experience with computers and basic training increased their confidence and reduced their resistance to embracing EHRs, which was essential for the effective implementation of the EHR system [60].

For continuous improvement of the use of EHRs, a training needs assessment is recommended to meet nurses’ current needs. One crucial issue that can affect nurses’ capacity to adapt and feel comfortable using EHRs is their level of prior computer knowledge. This gap can be closed and successful integration into EHR systems ensured by providing proper training and support for nurses who may have limited computer knowledge. In context, hospital authorities can identify the particular factors that limit nurses whilst using EHR. Addressing these issues with tailored programs can increase patient outcomes and patient safety.

### 4.1. Limitations

This study has several limitations. First, cross-sectional and longitudinal studies are recommended to provide more investigations about the factors that affect nurses’ attitude to EHRs. Second, this study used a convenience sample that is prone to biases and is less likely to represent the population. Finally, this study was conducted in Hail city hospitals and PHCs; future studies are needed to include large populations in Saudi Arabia to increase the generalizability of results.

### 4.2. Implications for Nursing Practice

The use of EHRs by the nursing profession has far-reaching consequences for the provision of healthcare and the treatment of patients. Leadership commitment and organizational support are essential for the successful implementation of an EHR system. Appropriate EHR-aligned resources, infrastructure, and policies must be put in place. Engaging nurses in decision-making processes and addressing their concerns can promote a favorable attitude toward EHR implementation. Nurses need to have access to dependable technical support to resolve any issues or obstacles they may encounter while using EHRs. To maintain nurses’ trust and confidence in EHRs, the system requires prompt resolution of technical problems. The user interfaces of EHR systems should be intuitive and simple to navigate. Complex or cumbersome interfaces can hinder the adoption and use of EHRs by nurses.

Nurses must have adequate training and education in EHR systems to use and navigate electronic platforms effectively, and this support should be provided so that nurses can improve their EHR proficiency and confidence. Increased efficiency, greater patient care, and better outcomes associated with EHRs can be realized in Saudi Arabian nursing practice by addressing these concerns and creating good attitudes toward EHR adoption.

## 5. Conclusions

The majority of the participating nurses had positive overall attitudes toward using EHRs. Nurses who are Saudi nationals, male, younger, have previous computer experience, a bachelor’s or diploma degree, and less than five years of experience were more likely to be those possessing overall positive attitudes. Moreover, sex, education level, and previous computer experience were independent factors of nurses’ knowledge of EHRs. Engaging nurses in decision-making processes and addressing their concerns can promote a favorable attitude toward EHR implementation. Optimizing the incorporation of EHRs into nursing practice is recommended; however, obstacles such as technology-related concerns and potential disturbances in nurse–patient relationships should be addressed as systemic problems affect proficiency with EHR.

## Figures and Tables

**Table 1 healthcare-11-02393-t001:** Sociodemographic and work-related characteristics of nurses (*n* = 297).

Characteristics		*n*	(%)
Sex			
	Male	93	(31.3)
	Female	204	(68.7)
Age	Mean ± SD	33.84 ± 6.61	
	≤30	92	(31.0)
	31–40	161	(54.2)
	≥40	44	(14.8)
Level of education			
	Diploma	70	(23.6)
	Bachelor	174	(58.6)
	Master	53	(17.8)
Nationality	Saudi	188	(63.3)
	Non-Saudi	109	(36.7)
Experience (years)	Mean ± SD	8.58 ± 5.73	
	Less than 5	85	(28.6)
	5–10	112	(37.7)
	More than 10	100	(33.7)
Previous computer experience			
	Yes	235	(79.1)
	No	62	(20.9)
Working unit			
	Inpatient units	62	(20.9)
	Outpatient units	42	(14.1)
	ICUs	107	(36.0)
	Others	86	(29.0)
Position			
	Supervising nurses	58	(19.5)
	Senior staff nurses	64	(21.5)
	Staff nurses	175	(58.9)

SD: standard deviation; ICUs: intensive care units.

**Table 2 healthcare-11-02393-t002:** Nurses’ attitudes toward questionnaire’s items.

#	Item	Strongly Disagree	Disagree	Uncertain	Agree	Strongly Agree	Mean ± SD
1	A computer increases costs by increasing the nurses’ workload.	5 (1.7)	66 (22.2)	80 (26.9)	111 (37.4)	35 (11.8)	3.35 ± 1.0
2	Computers cause a decrease in communication between hospital departments.	19 (6.4)	68 (22.9)	75 (25.3)	96 (32.3)	39 (13.1)	3.22 ± 1.1
3	Computers will allow the nurse more time for the professional tasks for which he or she is trained.	8 (2.7)	18 (6.1)	54 (18.2)	167 (56.2)	50 (16.8)	3.78 ± 0.88
4	Part of the increase in the costs of healthcare is because of computers.	7 (2.4)	27 (9.1)	76 (25.6)	142 (47.8)	45 (15.2)	3.64 ± 0.92
5	The time spent using a computer is out of proportion to the benefits.	6 (2.0)	49 (16.5)	84 (28.3)	124 (41.8)	34 (11.4)	3.44 ± 0.96
6	Computers represent a violation of patient privacy.	21 (7.1)	72 (24.2)	58 (19.5)	116 (39.1)	30 (10.1)	3.20 ± 1.13
7	Only one person at a time can use a computer terminal; therefore, staff efficiency is inhibited.	9 (3.0)	40 (13.5)	80 (26.9)	128 (43.1)	40 (13.5)	3.50 ± 0.98
8	The computerization of nursing data offers nurses a remarkable opportunity to improve patient care.	2 (0.7)	10 (3.4)	50 (16.8)	181 (60.9)	54 (18.2)	3.92 ± 0.73
9	Computers contain too much personal data to be used in an area as open as a nursing station.	6 (2.0)	30 (10.1)	68 (22.9)	148 (49.8)	45 (15.2)	3.65 ± 0.92
10	Computers cause nurses to give less time to quality patient care.	14 (4.7)	63 (21.2)	65 (21.9)	122 (41.1)	33 (11.1)	3.32 ± 1.07
11	If I had my way, nurses would never have to use computers.	41 (13.8)	86 (29.0)	51 (17.2)	85 (28.6)	34 (11.4)	2.94 ± 1.26
12	Computers should only be used in the financial department.	56 (18.9)	73 (24.6)	37 (12.5)	83 (27.9)	48 (16.2)	2.97 ± 1.38
13	Computers make nurses’ jobs easier.	4 (1.3)	18 (6.1)	43 (14.5)	157 (52.9)	75 (25.3)	3.94 ± 0.87
14	Paperwork for nurses has been greatly reduced by the use of computers.	5 (1.7)	15 (5.1)	38 (12.8)	155 (52.2)	84 (28.3)	4.00 ± 0.87
15	Orientation for new employees takes longer because of computers and, therefore, unnecessary work delays occur.	15 (5.1)	51 (17.2)	72 (24.2)	118 (39.7)	41 (13.8)	3.40 ± 1.08
16	Nursing information does not lend itself to computers.	16 (5.4)	41 (13.8)	78 (26.3)	131 (44.1)	31 (10.4)	3.40 ± 1.02
17	Computers save steps and allow the nursing staff to become more efficient.	4 (1.3)	8 (2.7)	52 (17.5)	167 (56.2)	66 (22.2)	3.95 ± 0.79
18	The more computers in an institution, the less the number of jobs for employees.	9 (3.0)	47 (15.8)	76 (25.6)	120 (40.4)	45 (15.2)	3.48 ± 1.02
19	Increased computer usage will allow nurses more time to give patient care.	7 (2.4)	28 (9.4)	67 (22.6)	136 (45.8)	59 (19.9)	3.71 ± 0.96
20	Because of computers, nurses will face more lawsuits.	10 (3.4)	48 (16.2)	82 (27.6)	121 (40.7)	36 (12.1)	3.42 ± 1.00

SD; standard deviation.

**Table 3 healthcare-11-02393-t003:** Relationship between nurses’ attitudes toward EHRs and their sociodemographic and work-related factors.

Variables		*n*	Mean Rank	*p*-Value
Sex ^a^				
	Male	93	173.94	<0.001 *
	Female	204	137.63	
Age ^b^				
	≤30	92	166.62	0.044 *
	31–40	161	143.52	
	≥40	44	132.19	
Previous computer experience ^a^				
	Yes	235	175.97	<0.001 *
	No	62	46.77	
Level of education ^b^				
	Diploma	70	122.79	0.003 *
	Bachelor	174	151.48	
	Master	53	175.47	
Nationality ^a^	Saudi	188	159.31	0.007 *
	Non-Saudi	109	131.22	
Experience (years) ^b^				
	Less than five	85	168.37	0.048 *
	5–10	112	141.78	
	More than 10	100	140.62	
Working unit ^b^				
	Inpatient units	62	157.79	0.215
	Outpatient units	42	163.94	
	ICU	107	135.97	
	others	86	151.58	
Position ^b^				
	Supervising nurses	58	154.78	0.071
	Senior staff nurses	64	127.16	
	Staff nurses	175	155.07	

* Significant; a, Mann–Whitney test; b, Kruskal–Wallis tests.

**Table 4 healthcare-11-02393-t004:** Multiple linear regression to determine the factors associated with nurses’ attitudes toward EHRs.

Factor		β	95% CI for β	*p*-Value
Sex				
	Male	Reference		
	Female	−2.874	−5.592–−0.157	0.038
Age		−0.075	−0.281–0.132	0.477
Level of education				
	Diploma	−6.667	−10.467–2.867	0.001
	Bachelor	−3.588	−6.928–0.248	0.035
	Master	Reference		
Nationality	Saudi	Reference		
	Non-Saudi	−1.864	−4.830–1.102	0.217
Previous computer	Yes	Reference		
	No	−16.588	−19.561–13.615	0.000

R^2^ = 0.368, adjusted R^2^ = 0.355, *p* < 0.001; CI, confidence interval.

## Data Availability

The data presented in this study are available on request from the corresponding author.

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
