# Peer review of "Nurses’ Attitudes and Factors Affecting Use of Electronic Health Record in Saudi Arabia"

_healthcare, 2023, doi:10.3390/healthcare11172393_

Round 1

Reviewer 1 Report

I have gone through the study and found it in good merit. The study highlighted the key perspectives about the Nurses’ attitudes and factors affecting the use of electronic health records in Saudi Arabia. but I have few concerns which should be addressed before the final acceptance of the study.

1. The abstract is too long for this study. It should be minimized for better understanding.

2. The methodology used in the study is not clear. Why the authors used convenience sampling. This is non-probability sampling which can create huge bias in the study. If the authors had to use non-probability sampling then there should be proper justification for it.

3. the author used the sampling calculator for calculating the sample size. the formula should be given by presenting the overall population. What is the overall population of nurses in Saudi Arabia? why did they choose the margin of error and on what basis they selected the margin of error? It all should be justified in a few lines.

3. Did the author use any scale for the knowledge questions? If yes then which scale and from which source did they choose the knowledge questions? please cite that reference.

4. there is no need to calculate the mean and SD of the qualitative questions. please mention and express them in numbers with percentages and interpret the results accordingly.

5. The authors used the Mann-Whitney test for calculating the p-values of qualitative variables which is not recommended in this case. Use binomial test or median test or other non-parametric test for nominal scale data.

6. Multiple linear regression is also not recommended in this situation. if your dependent variables are qualitative with two categories use binary logistic or more categories use multinomial regression and calculate the parameters accordingly.

7. I have serious concerns about the methodology of statistics used in the study which lacks knowledge of statistics and using the software to calculate the coefficients. The methodology and data analysis section should be consulted with an expert statistician.

8. results and discussion should be compared with other studies also. and discuss in which perspectives your study is better or not as compared to other studies.

the overall study is new and will add a good contribution to the literature but I have  concerns about the data analysis techniques applied in the study.  If concerns are answered accordingly , the study can be recommended.

Author Response

Response to Reviewers’ Comments

Dear Editor,

We would like to sincerely thank you for considering our manuscript for publication in Healthcare. We also gratefully thank the reviewers for critical and meticulous review that adds too much to the quality of our manuscript.

We have adhered to the reviewers’ comments and these responses outline how each comment was addressed. Changes to the manuscript are marked using track changes (referring to the corresponding Lines in the revised version), and a clean copy for the revised manuscript was also uploaded. Detailed responses to the points raised are as follows:

Reviewer #1

Comment:Comments and Suggestions for Authors

I have gone through the study and found it in good merit. The study highlighted the key perspectives about the Nurses’ attitudes and factors affecting the use of electronic health records in Saudi Arabia.”.

Response: Thank you for your support and encouragement.

Comment:but I have few concerns which should be addressed before the final acceptance of the study.”.

Response:

We appreciate your effort to improve our research. We appreciate your time and feedback. We have adhered to all your addressed comments one by one.

Comment:1. The abstract is too long for this study. It should be minimized for better understanding.”.

Response: Thank you for this comment. The abstract part was minimized to “359” words.

Comment:2. The methodology used in the study is not clear. Why the authors used convenience sampling. This is non-probability sampling which can create huge bias in the study. If the authors had to use non-probability sampling then there should be proper justification for it.”.

Response: Thank you for this comment. Despite the fact that the results of the Convenient Sample cannot be generalized due to the circumstances changing, the smaller community is the only one who can observe the results and outcomes that are relevant to this study. We added this statement “The nurses were selected by convenience sampling from the study population, one of the nonprobability sampling methods, as it is considered the most economical, easiest, as well as fastest way to collect data from the population (Aaker et. Al. 2007, Lecture rand Yaman, 2021)”. In addition, the authors considered it one of the limitations of the study that mentioned in the part of limitations of the study.

Aaker, D. A., Kumar, V., & Day, G. S. (2007). Marketing research (9th ed.). USA: John Wiley & Sons.

HoÅŸgör H, Yaman M. Investigation of the relationship between psychological resilience and job performance in Turkish nurses during the Covid-19 pandemic in terms of descriptive characteristics. J Nurs Manag. 2022 Jan;30(1):44-52. doi: 10.1111/jonm.13477. Epub 2021 Nov 10. PMID: 34595787; PMCID: PMC8646929.

Comment:3. the author used the sampling calculator for calculating the sample size. the formula should be given by presenting the overall population. What is the overall population of nurses in Saudi Arabia? why did they choose the margin of error and on what basis they selected the margin of error? It all should be justified in a few lines.”.

Response: Thank you for this comment. The total number of nurses who are working in the intended hospitals and PHCs are 1200 nurses. The margin of error was determined as 5% by default, and this was based on the expected uncertainty of 5% because the confidence level was 95%. This was described in Lines 130-136.

Comment:3. Did the author use any scale for the knowledge questions? If yes then which scale and from which source did they choose the knowledge questions? please cite that reference.”.

Response: Thank you for this comment. We did not involve knowledge questions. We used The Nurses’ Attitudes Towards Computerization (NATC) questionnaire that was developed by Stronge and Brodt (1985) to assess nurses’ attitudes toward EHRs. The details were described in the instrument part (Lines, 139-149).

Comment:4. there is no need to calculate the mean and SD of the qualitative questions. please mention and express them in numbers with percentages and interpret the results accordingly.”.

Response: Thank you for this comment. We would like to illustrate that we discussed the means of each item in the results and discussion parts that are revealed for the readers the result of each item rated by the nurses. To make the presentation of data more attractive, we integrate the columns of means and SD to one column. Please, if you want us to remove it, we will do that.

Comment:5. The authors used the Mann-Whitney test for calculating the p-values of qualitative variables which is not recommended in this case. Use binomial test or median test or other non-parametric test for nominal scale data.”.

Response:

Thank you for this comment. We would like to illustrate that the dependent variable of the study “attitudes” data was measured as the total scores (continuous) for each participant that ranged from 20-100 as described in the scoring system in the part of the instrument. For more illustration, we added this statement “nonparametric statistics (Mann–Whitney test or Kruskal–Wallis tests) were used in this study to investigate the relationship between the independent variables and the total scores of the attitudes.” to statistics part (Lines, 180 -182). 

Comment:6. Multiple linear regression is also not recommended in this situation. if your dependent variables are qualitative with two categories use binary logistic or more categories use multinomial regression and calculate the parameters accordingly.”.

Response:

Thank you for this comment. As illustrated in the previous comment we used the dependent variables as continuous variable, and it was illustrated for more details in the part of “Data analysis”.

Comment:7. I have serious concerns about the methodology of statistics used in the study which lacks knowledge of statistics and using the software to calculate the coefficients. The methodology and data analysis section should be consulted with an expert statistician.”.

Response:

Thank you for your comment. We appreciate your concern in this important point, and we would like to illustrate that we have a specialized statistician, and we discussed each point with him. Also, we hope that our illustrations regarding your concern were illustrated in the previous of your comments.

Comment:8. results and discussion should be compared with other studies also. and discuss in which perspectives your study is better or not as compared to other studies.”.

Response: Thank you for this comment. We discussed all the points arising in the results and compared them with the earlier studies and we added additional updated studies please find them in the discussion part with highlight color (Lines, 245-248, and 258-270).

Comment:the overall study is new and will add a good contribution to the literature, but I have concerns about the data analysis techniques applied in the study.  If concerns are answered accordingly, the study can be recommended.”.

Response: Thank you for your kindness and support. We hope all of your concerns were addressed. Thank you again for your effort to improve our manuscript.

We hope that we addressed the comments raised by the Reviewers, which contributed to the improvement of the quality of our manuscript. We hope that our revised manuscript is accepted for publication in Healthcare, and we are pleased to receive any further comments or suggestions. 

With kind regards,

Sameer A. Alkubati, PhD

The Corresponding Author

Email: s.alkubati@uoh.edu.sa

Reviewer 2 Report

Dear Editor/Authors,

Thank you for the opportunity to review this manuscript. Here are my comments for the manuscript in response to my recommendation.

Introduction

There is a clear overview of the topic, which is focused on nurses' attitudes toward Electronic Health Records (EHRs) and the factors influencing EHR implementation in hospitals in Saudi Arabia. It establishes the importance of EHRs in improving healthcare documentation, patient safety, and overall patient care.

Some comments to improve this section:

·        While the introduction mentions various claims about EHRs, their advantages, and their slow adoption in Saudi Arabia, it lacks proper citations and references to support these statements. Including relevant citations would enhance the article's credibility and allow readers to explore further.

·        Literature review needs to be more comprehensive to provide a broader context for the study, highlighting existing knowledge and identifying gaps that the study seeks to address.

Method:

This section is well-structured and provides sufficient information to understand the research methodology. Sample size is clearly achieved. Ethical considerations is addressed.

Some comments to improve this section:

·        L116 – L123: Formatting issue: Everything is italic. Please revise.

·        The study uses the NATC questionnaire with five domains but does not provide a detailed description of each domain. Including a brief explanation of the domains would help readers understand the specific aspects of nurses' attitudes being assessed.

·        Need to include the validity and reliability of the NATC questionnaire.  

·        A quick description of the hospitals is required to allow the international readers to under how significant is the study based on four hospitals and PHCs in Saudi Arabia.

·        How many PHCs were included?

·        Data collection process is described but there is little information about how potential biases or confounding factors were addressed during data collection.

 Result

The section is well organized and provides valuable data and highlights important factors influencing nurses' attitudes toward EHRs. No issue with the statistics here.

Discussion

The section is lengthy but comprehensive with adequate supporting evidence.

Some comments to improve this section

·        Please include the limitations of the study to provide a more balanced discussion of the results.

·        Last but not least, can the authors write how their study is unique and has contributed to the wider body of literature whereby there is quite a lot of studies in this area with similar results?

Overall, the English used in the article is understandable and conveys the main ideas effectively. However, there are a few areas where the language could be improved for clarity and coherence:

Author Response

Response to Reviewers’ Comments

Dear Editor,

We would like to sincerely thank you for considering our manuscript for publication in Healthcare. We also gratefully thank the reviewers for critical and meticulous review that adds too much to the quality of our manuscript.

We have adhered to the reviewers’ comments and these responses outline how each comment was addressed. Changes to the manuscript are marked using track changes (referring to the corresponding Lines in the revised version), and a clean copy for the revised manuscript was also uploaded. Detailed responses to the points raised are as follows:

Reviewer #2

Comments and Suggestions for Authors

Dear Editor/Authors,

Comment:Thank you for the opportunity to review this manuscript. Here are my comments for the manuscript in response to my recommendation.”.

 Response:

Thank you for your effort to improve our research. We appreciate your time and feedback. We have adhered to all your addressed comments one by one.

Introduction

Comment:There is a clear overview of the topic, which is focused on nurses' attitudes toward Electronic Health Records (EHRs) and the factors influencing EHR implementation in hospitals in Saudi Arabia. It establishes the importance of EHRs in improving healthcare documentation, patient safety, and overall patient care.”.

Response:

Thank you for your kindness and support.

Comment:Some comments to improve this section:”.

  • Response: We have adhered to all your addressed comments one by one.

       Comment:While the introduction mentions various claims about EHRs, their advantages, and their slow adoption in Saudi Arabia, it lacks proper citations and references to support these statements. Including relevant citations would enhance the article's credibility and allow readers to explore further.”.

Response:

Thank you for bringing this important note up. We rephrased the introduction part and added more updated and relevant references to it (References No. 4, 5, 6, 8, 11, 13, 16, 17, 20, 21)

Comment:Literature review needs to be more comprehensive to provide a broader context for the study, highlighting existing knowledge and identifying gaps that the study seeks to address.”.

Response: Thank you for this comment. We improved it and added the existing knowledge and the gap of the study. Please see the highlighted paragraph in the introduction section (Lines, 94-101, and 107-110).

Method:

Comment:This section is well-structured and provides sufficient information to understand the research methodology. Sample size is clearly achieved. Ethical considerations is addressed.”.

Response:

Thank you for your kindness and support.

Some comments to improve this section:

  • Comment:L116 – L123: Formatting issue: Everything is italic. Please revise.”.

Response: Thank you for this note. We corrected it.

Comment:The study uses the NATC questionnaire with five domains but does not provide a detailed description of each domain. Including a brief explanation of the domains would help readers understand the specific aspects of nurses' attitudes being assessed.”.

Response: Thank you for this comment. We included more details and explanation related to the domains and the questionnaire.

Comment:Need to include the validity and reliability of the NATC questionnaire.”.  

Response:

Thank you for this comment. Validity and reliability of the NATC questionnaire were discussed in the part of “Instrument”, please you can find it with highlight color (Lines 151-156).     

Comment:A quick description of the hospitals is required to allow the international readers to under how significant is the study based on four hospitals and PHCs in Saudi Arabia.”.

Response:

Thank you for this comment. We added a description for the hospitals in the part of “Design and setting”.

  • Comment:How many PHCs were included?”

Response: Thank you for this question. The PHCs in Hail city are 28 center that were included in the study.

Comment:Data collection process is described but there is little information about how potential biases or confounding factors were addressed during data collection.”.

Response: Thank you for this comment. In this current study, the researchers minimized the effect of confounding variables by way employing statistical methodology which is the multiple regression.

 Result

Comment:The section is well organized and provides valuable data and highlights important factors influencing nurses' attitudes toward EHRs. No issue with the statistics here.”.

 Response: Thank you for your kindness and support.

Discussion

Comment:The section is lengthy but comprehensive with adequate supporting evidence.”.

Response: Thank you for your kindness and support.

Some comments to improve this section

  • Comment:Please include the limitations of the study to provide a more balanced discussion of the results.”.

Response: Thank you for bringing this important note up. We added the part of limitations at the end of discussion part (Lines, 429-435)

  • Comment:Last but not least, can the authors write how their study is unique and has contributed to the wider body of literature whereby there is quite a lot of studies in this area with similar results?”.

Response: Thank you for this comment. Our study results concentrated on the nurses who are considered the main healthcare providers who deal with patients for more periods of time and their improvements skills for using EHRs is considered very important issue to improve patients’ safety. Our results may give considerations about the importance of give more attention to improve nurses’ skills and attitudes toward EHRs. Also, the factors that arise from our study that had impact on the nurses’ attitude may be considered for future research and educational programmes interventions. We considered these points at the end of the discussion part (Lines, 422-426) .

Comments on the Quality of English Language

Comment:Overall, the English used in the article is understandable and conveys the main ideas effectively. However, there are a few areas where the language could be improved for clarity and coherence:”.

Response: Thank you for your kindness and support. We managed and improved it.

We hope that we addressed the comments raised by the Reviewers, which contributed to the improvement of the quality of our manuscript. We hope that our revised manuscript is accepted for publication in Healthcare, and we are pleased to receive any further comments or suggestions. 

With kind regards,

Sameer A. Alkubati, PhD

The Corresponding Author

Email: s.alkubati@uoh.edu.sa

Reviewer 3 Report

This is a study of nurses attitudes regarding electronic health records (EHRs) in Saudi Arabia. The authors found that younger, male, Saudi national nurses who have master’s degrees are the most likely to have positive attitudes regarding EHRs and older, female, foreign-trained nurses with less education the least likely. The authors found that there were systemic problems with the ease of adoption of EHRs, yet their recommendations most decidedly point to the need for changes in curricula—this even though this study collected no data on curricula and the current curricula is producing the type of nurses who are positive towards EHRs.

The authors make a number of unsupported claims and have not provided sufficient information on their methods to make it clear why they chose them. Furthermore, they have neglected to note the importance of one of their results and have not included a limitations section.

The authors need to look closely at their data and make their recommendations based only on what their data demonstrate rather than calling for changes in curricula that are unsubstantiated by their findings.

Line by line suggested edits.

66 Please included a supporting reference for the claim that EHRs were first implemented in Saudi Arabia a few decades ago.

70 Change “at lot” to “significant”.

70-73 Change “A patient’s medical history is represented electronically and may include crucial administrative, clinical, laboratory, and radiological data [9].It also makes it easier for different healthcare professionals to save and share priceless health data.” to “Once a patient’s medical history is represented electronically it may include crucial administrative, clinical, laboratory, and radiological data [9], making it easier for different healthcare professionals to save and share valuable health data.”

84 Change “avant-garde” to “proactive”.

88 Change “low” to “slow”.

91 Change “adverse events” to “challenges”.

94 Please state who this research is intended to serve and provide a summary of the major findings.

96 Please state why a cross-sectional study was selected and provide a current supporting reference to identify the use of cross-sectional studies in conducting similar research.

100 Delete “conveniently”

101 Please state why OpenEpi was selected and provide a current reference showing that similar research has also used OpenEpi.

107 Please explain why this questionnaire was selected among others and provide a current supporting reference for using this questionnaire.

116-123 Please remove the italics from these paragraphs.

116 Please explain how the nurses were invited plus the inclusion and exclusion criteria for participation.

123 Please state what the incentive was for the nurses to participate in the research study.

127 Change “ was given” to “signed a”.

131 Please explain why this version of SPSS was used for the analysis and provide a current supporting reference for using this software for analyzing data for similar studies.

134 Please provide a reference for the Kolmogorov-Smimov test.

137 Please provide a reference for both the Mann-Whitney test and the Kruskal-Wells tests.

147 Note that there are no data for “Senior staff nurses”.

158 The authors should note that “Computers save steps and allow the nursing staff to become more efficient” has a mean of 3.95.

162 Please indicate the content of each of the columns in the heading.

179 Please center “Variables” over both the first to columns similar to how “Factor” is centered in Table 4.

182 Please be clear on whether the nurses were tested for gender or for sex. The text says “gender” when the table refers to “sex”.

184 Table 4 has no heading. Please provide one. 

194 Change “as indicated in” to “similar to”.

200 Change “not thrilled” to “unsatisfied”.

206 Change “an environment” to “their working conditions”.

206-212 This part of the paragraph does not follow from the beginning sentences. The beginning part points to problems in the implementation of EHRs being system-related. This second part of the paragraph instead stresses what nurses need to do rather than indicating how systems should be modified to improve the implementation of EHRs. Please change the second part of the paragraph to discuss the changes that should be made to the system.

215 “Computers save steps and allow the nursing staff to become more efficient” was higher than “Computers make nurses’ jobs easier” and “The computerization of nursing data offers nurses a remarkable opportunity to improve patient care”. Please note this.

233-238 There are three separate claims made in each of these three sentences. Each requires a current supporting reference.

238-242 If younger nurses are more likely to be competent regarding the use of EHRs, why should nursing schools make it a top priority to teach students how to use computers and the internet effectively? It seems that younger nurses are gaining the right type of education currently. Furthermore, given that the authors did not examine pedagogical practices, they can only draw conclusions based on their data. As such, they cannot make these type of recommendations regarding pedagogy.

244-245 Sex and gender do not mean the same things. They cannot be used interchangeably. Choose one or the other. This will depend on whether the authors tested for sex (that is whether the participants had an x or a y chromosome) or they tested for gender (how the participants present themselves regarding sexual norms).

269-270 What about the data collected leads the authors to conclude that nursing curricula need to be improved? If the younger nurses are competent in the use of EHRs then why do the authors think it necessary to change the curricula? Please explain.

284 Why do the authors say their study shows that all nurses need EHR-specific training? It seems that those who are younger with master’s degrees already are receiving sufficient training. Please explain.

288 The authors have not studied the nursing curriculum. As such they cannot make these type of claims regarding how the nursing curriculum should be updated. This is especially so since the current training nursing are receiving appears sufficient from the data. The problem seems to be with the older nurses and ones who are not trained in the country. Please take this into consideration and leave out this recommendation.

292 Why are the authors recommending the development of advanced nursing degrees based on their data collected rather than merely suggesting that more nurses take advanced nursing degrees? What of the data collected makes the authors believe that current pedagogy is the problem? Please explain.

296 The authors should note that not only are the newer nurses more responsive and flexible, they have been better trained.

324 Again, why are the authors recommending individualized education programs for the nurses in this regard? What of their data collected leads them to make this recommendation? Wouldn’t it be more reasonable to recommend that older nurses who don’t feel comfortable with EHRs take professional development courses to update their skills regarding EHRs? Please comment on this.

341 What evidence do the authors have that specialized education programs need to be created? It appears that the younger nurses are proficient with EHRs, it is the ones who are older who are not. These nurses need professional development. Please explain why the authors think that the creation of specialized programs is necessary.

343-345 The younger nurses and those with master’s degrees do have this prior computer knowledge. What proper training are they missing? The authors are asked to explain.

345 From the data collected, it appears that the problem lies both with how systems are employing EHRs and the difference between the skills of the younger, better educated nurses and the skills of the older nurses that are not trained in the country. Therefore, it is not nursing education that needs to change, it is practices. Please comment.

356 Changed “HER” to “EHR”.

 372 Change “experience.” to “experience were more likely to be those possessing positive overall attitudes.”.

375-377 This recommendation regarding changes to nursing education has not been justified by the data collected. Not only did the authors not test pedagogy, the data they collected demonstrated that current education practices were of the type necessary for nurses to be positive towards the use of EHRs. Please modify the conclusion as a result.

378 What the authors have not mentioned or investigated in the literature is how systemic problems affect proficiency with EHR. This needs further investigation and mention in the conclusions.

The English is fine. There are only a few changes to be made that have been specified in the Comments and Suggestions for Authors.

Author Response

Response to Reviewers’ Comments

Dear Editor,

We would like to sincerely thank you for considering our manuscript for publication in Healthcare. We also gratefully thank the reviewers for critical and meticulous review that adds too much to the quality of our manuscript.

We have adhered to the reviewers’ comments and these responses outline how each comment was addressed. Changes to the manuscript are marked using track changes (referring to the corresponding Lines in the revised version), and a clean copy for the revised manuscript was also uploaded. Detailed responses to the points raised are as follows:

Reviewer #3

Comments and Suggestions for Authors

Comment:This is a study of nurses attitudes regarding electronic health records (EHRs) in Saudi Arabia. The authors found that younger, male, Saudi national nurses who have master’s degrees are the most likely to have positive attitudes regarding EHRs and older, female, foreign-trained nurses with less education the least likely. The authors found that there were systemic problems with the ease of adoption of EHRs, yet their recommendations most decidedly point to the need for changes in curricula—this even though this study collected no data on curricula and the current curricula is producing the type of nurses who are positive towards EHRs.”.

Response: Thank you for this comment. We removed the statements that related to the curricula.

Comment:The authors make a number of unsupported claims and have not provided sufficient information on their methods to make it clear why they chose them. Furthermore, they have neglected to note the importance of one of their results and have not included a limitations section.”.

Response: Thank you for this comment. The introductions, methodology, discussion parts of the manuscript were more improved, and the part of limitations section was added at the end of discussion.

Comment:The authors need to look closely at their data and make their recommendations based only on what their data demonstrate rather than calling for changes in curricula that are unsubstantiated by their findings.”.

 Response: Thank you for this comment. This has been clarified at the conclusion section and we removed the statements that related to the curricula (Lines, 464-467).

Line by line suggested edits.

Comment:66 Please included a supporting reference for the claim that EHRs were first implemented in Saudi Arabia a few decades ago.”.

Response: Thank you for this comment. The reference was provided, and all the statements of the introduction part were provided with updated references.

Comment:70 Change “at lot” to “significant”.”.

Response: Thank you for this note. It was changed.

Comment:70-73 Change “A patient’s medical history is represented electronically and may include crucial administrative, clinical, laboratory, and radiological data [9].It also makes it easier for different healthcare professionals to save and share priceless health data.” to “Once a patient’s medical history is represented electronically it may include crucial administrative, clinical, laboratory, and radiological data [9], making it easier for different healthcare professionals to save and share valuable health data.”

Response: Thank you for this note. It was replaced.

Comment:84 Change “avant-garde” to “proactive”.

Response: Thank you for this note. It was changed.

Comment:88 Change “low” to “slow”.

Response: Thank you for this note. It was changed.

Comment:91 Change “adverse events” to “challenges”.

Response: Thank you for this note. It was changed.

Comment:94 Please state who this research is intended to serve and provide a summary of the major findings.”.

Response: Thank you for this comment. This part of Introduction was improved by extending more literatures (Lines, 94-101). In addition, we added statements to describe who research is intended to serve at the end of introduction part (lines, 106-110)

Comment:96 Please state why a cross-sectional study was selected and provide a current supporting reference to identify the use of cross-sectional studies in conducting similar research.”.

Response: Thank you for this point. We provided an explanation with reference in the beginning of the “Design and setting” part.

Comment:100 Delete “conveniently

Response: Thank you for this note. It was deleted.

Comment:101 Please state why OpenEpi was selected and provide a current reference showing that similar research has also used OpenEpi.”.

Response: Thank you for this comment. We provided an explanation with reference in the sample section.

Comment:107 Please explain why this questionnaire was selected among others and provide a current supporting reference for using this questionnaire.”.

Response: Thank you for this point. We provided explanation about the reasons to use this instrument with related references in the part of instrument (Lines, 139-149).

Comment:116-123 Please remove the italics from these paragraphs.”.

Response: Thank you for this note. It was corrected.

Comment:116 Please explain how the nurses were invited plus the inclusion and exclusion criteria for participation.”

Response: Thank you for this comment. The invitation of nurses was described in the Data collection part (Lines, 158-162). The inclusion and exclusion criteria were described in the sample part (Lines, 126-130)

Comment:123 Please state what the incentive was for the nurses to participate in the research study.”.

Response: Thank you for this comment. The nurses did not receive any incentive to participate in the study. We added a statement to show that in the section of “Ethical Considerations” (Lines, 171-172).

Comment:127 Change “was given” to “signed a”.

Response: Thank you for this note. It was changed.

Comment:131 Please explain why this version of SPSS was used for the analysis and provide a current supporting reference for using this software for analyzing data for similar studies.”.

Response: Thank you for this comment. We used this version because it was the most updated and available version for us during the data analysis. The references for this version and studies used it were provided in the data analysis part.

Comment:134 Please provide a reference for the Kolmogorov-Smimov test.”

Response: Thank you for this point. We provided the reference for it.

Comment:137 Please provide a reference for both the Mann-Whitney test and the Kruskal-Wells tests.”.

Response: Thank you for this point. We provided the reference for them.

Comment:147 Note that there are no data for “Senior staff nurses”.

Response: Thank you for this important note. We added their data in the table.

Comment:158 The authors should note that “Computers save steps and allow the nursing staff to become more efficient” has a mean of 3.95.”.

Response: Thank you for bringing this important point up. The statement was corrected, and this item was added (Lines. 200-201).

Comment:162 Please indicate the content of each of the columns in the heading.”

Response: Thank you for this suggestion. We indicated each of the column in the heading of the table.

Comment:179 Please center “Variables” over both the first to columns similar to how “Factor” is centered in Table 4.”

Response: Thank you for this note. It was centered.

Comment:182 Please be clear on whether the nurses were tested for gender or for sex. The text says “gender” when the table refers to “sex”.”.

Response: Thank you for this note. We changed the gender to sex.

Comment:184 Table 4 has no heading. Please provide one.”. 

Response: Thank you and sorry for this mistake. We added the title of the table 4.

Comment:194 Change “as indicated in” to “similar to”.

Response: Thank you for this comment. We changed it.

Comment:200 Change “not thrilled” to “unsatisfied”.

Response: Thank you for this comment. We relaced it.

Comment:206 Change “an environment” to “their working conditions”.

Response: Thank you for this comment. We changed it.

Comment:206-212 This part of the paragraph does not follow from the beginning sentences. The beginning part points to problems in the implementation of EHRs being system-related. This second part of the paragraph instead stresses what nurses need to do rather than indicating how systems should be modified to improve the implementation of EHRs. Please change the second part of the paragraph to discuss the changes that should be made to the system.”.

Response: Thank you very much for pointing this out. We have changed this particular discussion as recommended. Please see highlighted sentences in this particular paragraph (Lines, 258-263).

Comment:215 “Computers save steps and allow the nursing staff to become more efficient” was higher than “Computers make nurses’ jobs easier” and “The computerization of nursing data offers nurses a remarkable opportunity to improve patient care”. Please note this.”

Response: Thank you for this comment. We changed it in results part and discussion (Line 278).

Comment:233-238 There are three separate claims made in each of these three sentences. Each requires a current supporting reference.”

Response: Thank you for this comment. Each sentence was provided with relevant reference (Lines, 296-302).

Comment:238-242 If younger nurses are more likely to be competent regarding the use of EHRs, why should nursing schools make it a top priority to teach students how to use computers and the internet effectively? It seems that younger nurses are gaining the right type of education currently. Furthermore, given that the authors did not examine pedagogical practices, they can only draw conclusions based on their data. As such, they cannot make these types of recommendations regarding pedagogy.”.

Response: Thank you very much. We removed the sentences pertaining to the recommendation regarding pedagogy.

Comment:244-245 Sex and gender do not mean the same things. They cannot be used interchangeably. Choose one or the other. This will depend on whether the authors tested for sex (that is whether the participants had an x or a y chromosome) or they tested for gender (how the participants present themselves regarding sexual norms).”.

Response: Thank you for this important point. We particularly changed the gender to sex.

Comment:269-270 What about the data collected leads the authors to conclude that nursing curricula need to be improved? If the younger nurses are competent in the use of EHRs then why do the authors think it necessary to change the curricula? Please explain.”.

Response: Thank you for this comment. We made a clarification on the statement “Additionally, utilising these people's positive attitudes and current digital competencies might help nursing administrators to understand training of EHR as early as possible ensures that future nurses are ready to accept and use technology in their professional positions”, please find it with highlight color (Lines, 340-243).

Comment:284 Why do the authors say their study shows that all nurses need EHR-specific training? It seems that those who are younger with master’s degrees already are receiving sufficient training. Please explain.”.

Response: Thank you for this note. We rephrase it to the following “This study shows that nurses, especially those older and without master’s degrees, need EHR-specific training and instruction. These training programs should train nurses to make better use of EHRs by making them more familiar with their features and benefits.” To make more concentration for older nurses and without master’s degrees (Lines, 358-361).

Comment:288 The authors have not studied the nursing curriculum. As such they cannot make these type of claims regarding how the nursing curriculum should be updated. This is especially so since the current training nursing are receiving appears sufficient from the data. The problem seems to be with the older nurses and ones who are not trained in the country. Please take this into consideration and leave out this recommendation.”.

Response: Thank you. We removed the sentences pertaining to nursing curricula.

Comment:292 Why are the authors recommending the development of advanced nursing degrees based on their data collected rather than merely suggesting that more nurses take advanced nursing degrees? What of the data collected makes the authors believe that current pedagogy is the problem? Please explain.”.

Response: Although, while we strike out this sentence in this revision we just want to put this claim in perspective that the current pedagogy needs to devote time to train the future workforce in EHR. However, we removed this sentence and rephrased the paragraph.

Comment:296 The authors should note that not only are the newer nurses more responsive and flexible, they have been better trained.”

Response: Thank you this is comment. It was noted in the paragraph (Line, 371).

Comment:324 Again, why are the authors recommending individualized education programs for the nurses in this regard? What of their data collected leads them to make this recommendation? Wouldn’t it be more reasonable to recommend that older nurses who don’t feel comfortable with EHRs take professional development courses to update their skills regarding EHRs? Please comment on this.”.

Response: Thank you. Since we did not look to explore the pedagogical data, this particular sentence is removed.

Comment:341 What evidence do the authors have that specialized education programs need to be created? It appears that the younger nurses are proficient with EHRs, it is the ones who are older who are not. These nurses need professional development. Please explain why the authors think that the creation of specialized programs is necessary.”

Response: We rephrased this statement in this particular paragraph to “ For continuous improvement of the EHR use, a training needs assessment is recommended to meet nurses’ current needs.” (Lines, 416-417).

Comment:343-345 The younger nurses and those with master’s degrees do have this prior computer knowledge. What proper training are they missing? The authors are asked to explain.”

Response: We clarify this sentence as there is no exact data about the training they need. We made mentioned the recommendation of training needs to assess the current needs of the nurses (Line, 419).

Comment:345 From the data collected, it appears that the problem lies both with how systems are employing EHRs and the difference between the skills of the younger, better educated nurses and the skills of the older nurses that are not trained in the country. Therefore, it is not nursing education that needs to change, it is practices. Please comment.”.

Response: While we strike this out (to focus only on the data), we believe that both the practice and nursing education may be the problems. As observed in nursing education, there is a lack of time devoted to training students on EHR.

Comment:356 Changed “HER” to “EHR”.

Response: Thank you for this note. We changed it.

 Comment:372 Change “experience.” to “experience were more likely to be those possessing positive overall attitudes.”.

Response: Thank you for this important comment. We changed it.

Comment:375-377 This recommendation regarding changes to nursing education has not been justified by the data collected. Not only did the authors not test pedagogy, the data they collected demonstrated that current education practices were of the type necessary for nurses to be positive towards the use of EHRs. Please modify the conclusion as a result.”.

Response: Thank you. We removed the nursing education to be consistent and rephrased the sentence.

Comment:378 What the authors have not mentioned or investigated in the literature is how systemic problems affect proficiency with EHR. This needs further investigation and mention in the conclusions.”.

Response: Thank you. We added this as part of the conclusion.

Comments on the Quality of English Language

Comment:The English is fine. There are only a few changes to be made that have been specified in the Comments and Suggestions for Authors.”

Response: Again, thank you for your effort and time to improve our manuscript.

We hope that we addressed the comments raised by the Reviewers, which contributed to the improvement of the quality of our manuscript. We hope that our revised manuscript is accepted for publication in Healthcare, and we are pleased to receive any further comments or suggestions. 

With kind regards,

Sameer A. Alkubati, PhD

The Corresponding Author

Email: s.alkubati@uoh.edu.sa

Round 2

Reviewer 1 Report

The authors have addressed all the issues raised in the review. 

Author Response

Comment: The authors have addressed all the issues raised in the review

Response: Thank you for your support and encouragement. We really appreciate your effort to improve our research. We appreciate your time and feedback that improved our manuscript.

With kind regards,

Sameer A. Alkubati, PhD

The Corresponding Author

Email: s.alkubati@uoh.edu.sa

Reviewer 3 Report

The authors are thanked for the changes they have made to their manuscript based on the suggestions of this reviewer. These changes have improved the paper. There remain some points that require clarification. Although the authors no longer are making recommendations regarding curricula, as per the advice of this reviewer, they do make other recommendations that are still not based on the data they collected. These parts of the paper require improvement.

The authors also should note that research regarding electronic health records is continually evolving. As such, references cited need to be current, that is, within the last five years. A number of the references are older than 2018. If the authors want to use these references they also have to provide additional references that are from 2018 at the earliest. Please go through the paper and either substitute later references for those before 2018 or add supporting  references no older than from 2018. 

For citations, if more than one article is cited, eliminate the space between the article number and the comma.

Line by line suggested edits.

88 Change “study demonstrated” to “, a study demonstrated”.

91 Change “Arabia such as inadequate knowledge in computer, a” to “Arabia, such as inadequate computer knowledge, a”.

122-123 Change “However, 400” to “400”.

188-189 Table 2

Please put the title “Strongly” and “agree” on two lines. This will decrease the size of that column, permitting the closing bracket to be on the same line as the rest of the information for each entry. 

For item 2, the spacing for the two right-most entries is off from the rest of the column for each. Please line up these entries in the columns.

224 Change “Additionally, a” to “Additionally,”.

231 A 2016 study cannot be the only reference used to claim that nurses are unsatisfied. They may have been unsatisfied in 2016, but that was seven years ago. Please find a recent reference in a peer reviewed journal to support this claim or change “Nurses are unsatisfied” to “In 2016, nurses were found unsatisfied”.

244 “working with the technology industry”—what do the authors mean by this? How would this be accomplished and why would it require working with the technology industry for nurses to understand how to use common software? The authors are asked to please explain this advice they have offered.

245 “A better selection process” the authors have not tested the selection process, so they can’t make recommendations regarding bettering this process.

247-250 Although each of the points made in this sentence seems reasonable, the authors did not test for what makes an EHR application successful. Therefore, they cannot make recommendations regarding the points they have made in this sentence.

279-280 “Nurses’ willingness to adopt computerization is indicative of their understanding of the merits of using technology in their profession [41].”—the only difference between this statement and that made in lines 272-274 is that this second time the point is mentioned, the authors have provided a reference. Please eliminate one of these sentences.

293 Delete “naturally”—the authors only know that men showed they were more comfortable and enthusiastic, they did not test if this was a result of nature or nurture.

325 If the authors are wanting to reference a 2015 study then change “Middle East have seen” to “Middle East saw”.

382-384 “The uniqueness of this study is that the data were taken from the main hospitals of one of the biggest regions of Saudi Arabia.”—this information belongs as the last line of the Introduction, not here. Please move this to the end of line 102.

References

Please check the Instructions for Authors of Healthcare regarding the appropriate method for referencing. Titles of articles and journal numbers should not be in bold—year of publication should. Issue numbers should be eliminated and DOI numbers included. Furthermore, the shortened journal names should have a period after each contraction and there should be a comma between the journal number and page numbers—not a colon.

Reference 56 is not complete. The journal name is missing.

There are only a few small things to changes regarding the English. These are mentioned in the Comments and Suggestions for Authors.

Author Response

Response to Reviewers’ Comments

Dear Editor,

We would like to sincerely thank you for considering our manuscript for publication in Healthcare. We also gratefully thank the reviewers for critical and meticulous review that adds too much to the quality of our manuscript.

We have adhered to the reviewers’ comments and these responses outline how each comment was addressed. Changes to the manuscript are marked using track changes (referring to the corresponding Lines in the revised version), and a clean copy for the revised manuscript was also uploaded. Detailed responses to the points raised are as follows:

Reviewer #3

Comments and Suggestions for Authors

Comment: The authors are thanked for the changes they have made to their manuscript based on the suggestions of this reviewer. These changes have improved the paper.

Response: Thank you for your support and encouragement. We really appreciate your effort to improve our research. We appreciate your time and feedback.

Comment: There remain some points that require clarification.

Response: Thank you for your points. We adhered to clarify your points one by one.

Comment: Although the authors no longer are making recommendations regarding curricula, as per the advice of this reviewer, they do make other recommendations that are still not based on the data they collected. These parts of the paper require improvement.

Response: Thank you for this comment. We had deleted recommendations that are not based on the data as suggested. Please see changes in the discussion part and conclusion.

Comment: The authors also should note that research regarding electronic health records is continually evolving. As such, references cited need to be current, that is, within the last five years. A number of the references are older than 2018. If the authors want to use these references they also have to provide additional references that are from 2018 at the earliest. Please go through the paper and either substitute later references for those before 2018 or add supporting references no older than from 2018. 

Response: Thank you for bringing up this important note. We replaced the old references and supported some of them with updated references. Just references that related to the validity and reliability of questionnaire as their importance.

Comment: For citations, if more than one article is cited, eliminate the space between the article number and the comma.

Response: Thank you for this note. We revised all these references citations, and we provided a clean version without Endnote, that is more accurate, and we adhered to the Healthcare journal guidelines.

Line by line suggested edits.

Comment: 88 Change “study demonstrated” to “, a study demonstrated”.

Response: Thank you for this note. It was changed.

Comment: 91 Change “Arabia such as inadequate knowledge in computer, a” to “Arabia, such as inadequate computer knowledge, a”.

Response: Thank you for this comment. We changed it.

Comment: 122-123 Change “However, 400” to “400”.

Response: Thank you for this comment. We changed it.

 Comment: 188-189 Table 2 Please put the title “Strongly” and “agree” on two lines. This will decrease the size of that column, permitting the closing bracket to be on the same line as the rest of the information for each entry.

Response: Thank you for this note. It was done. Please, also if you can see the clean version because it may be related to a problem with word.

Comment: For item 2, the spacing for the two right-most entries is off from the rest of the column for each. Please line up these entries in the columns.

Response: Thank you for this note. It was corrected in the clean version, we adhered to the guidelines of Healthcare Journal.  The highlight version with endnote and we couldn’t modify it due to technical problem. Also, we couldn’t change the tracking version without Endnote because the form of journal word is changing. So, we corrected the reference on the clean version of the manuscript.

Comment: 224 Change “Additionally, a” to “Additionally,”.

Response: Thank you for this note. It was changed.

 Comment: 231 A 2016 study cannot be the only reference used to claim that nurses are unsatisfied. They may have been unsatisfied in 2016, but that was seven years ago. Please find a recent reference in a peer reviewed journal to support this claim or change “Nurses are unsatisfied” to “In 2016, nurses were found unsatisfied”.

Response: Thank you for this note. We changed it. As well as this reference was supported by references in 2023 and 2020.

Comment: 244 “working with the technology industry”—what do the authors mean by this? How would this be accomplished and why would it require working with the technology industry for nurses to understand how to use common software? The authors are asked to please explain this advice they have offered.

Response: Thank you for this comment. We would like to put into perspective that working with the technology industry means that while nursing judgment based on critical thinking and decision-making is supported and informed by EHR, nurses must continually advance their computer skills through training by people in the IT healthcare field. By way of working together. nurses can also recommend programmes or applications that are essential to their profession. Although, we decided to strike this paragraph in relation to succeeding comments made.

 Comment: 245 “A better selection process” the authors have not tested the selection process, so they can’t make recommendations regarding bettering this process.

Response: Thank you for this note. We removed this line.

Comment: 247-250 Although each of the points made in this sentence seems reasonable, the authors did not test for what makes an EHR application successful. Therefore, they cannot make recommendations regarding the points they have made in this sentence.

Response: Thank you. We strike this out.

Comment: 279-280 “Nurses’ willingness to adopt computerization is indicative of their understanding of the merits of using technology in their profession [41].”—the only difference between this statement and that made in lines 272-274 is that this second time the point is mentioned, the authors have provided a reference. Please eliminate one of these sentences.

Response: Thank you for this important note. We removed the second.

Comment: 293 Delete “naturally”—the authors only know that men showed they were more comfortable and enthusiastic, they did not test if this was a result of nature or nurture.

Response: Thank you for this comment. We removed it.

Comment: 325 If the authors are wanting to reference a 2015 study then change “Middle East have seen” to “Middle East saw”.

Response: Thank you for this comment. It was changed.

Comment: 382-384 “The uniqueness of this study is that the data were taken from the main hospitals of one of the biggest regions of Saudi Arabia.”—this information belongs as the last line of the Introduction, not here. Please move this to the end of line 102.

Response: Thank you for this comment. It was transferred.

References

Comment: Please check the Instructions for Authors of Healthcare regarding the appropriate method for referencing. Titles of articles and journal numbers should not be in bold—year of publication should. Issue numbers should be eliminated and DOI numbers included. Furthermore, the shortened journal names should have a period after each contraction and there should be a comma between the journal number and page numbers—not a colon.

Response: Thank you for this important point. We revised all these references citations, and we provided a clean version without Endnote, because we couldn’t modify the highlight version that is with Endnote., that is more accurate, and we adhered to the Healthcare journal guidelines.

Comment: Reference 56 is not complete. The journal name is missing.

Response: Thank you for this comment. We corrected it accordingly and revised all the references. Please, see the clean version without Endnote.

Comments on the Quality of English Language

Comment: There are only a few small things to changes regarding the English. These are mentioned in the Comments and Suggestions for Authors.

Response: Again, we appreciate your effort and time regarding your comments that improved our paper. We hope that we adhered and corrected all your comments.

We hope that we addressed the comments raised by the Reviewers, which contributed to the improvement of the quality of our manuscript. We hope that our revised manuscript is accepted for publication in Healthcare, and we are pleased to receive any further comments or suggestions. 

With kind regards,

Sameer A. Alkubati, PhD

The Corresponding Author

Email: s.alkubati@uoh.edu.sa
